# Effect of 3-Nitropropionic Acid at Different Doses on In Vitro Rumen Fermentation, Digestibility, and Methane Emissions of Grazing Yak and Cattle

**DOI:** 10.3390/ani14121804

**Published:** 2024-06-17

**Authors:** Wei Guo, Weiwei Wang, Ying Zhang, Mi Zhou

**Affiliations:** 1Key Laboratory of Animal Genetics, Breeding and Reproduction in the Plateau Mountainous Region, Ministry of Education, Guizhou University, Guiyang 550025, China; wguo@gzu.edu.cn (W.G.); guowei6866@gmail.com (W.W.); 2Department of Agricultural, Food and Nutritional Science, University of Alberta, Edmonton, AB T6G 2P5, Canada; 3School of Public Health, Lanzhou University, Lanzhou 730000, China; yingz@lzu.edu.cn

**Keywords:** in vitro gas production, rumen fermentation, cattle and yak, 3-nitropropionic acid

## Abstract

**Simple Summary:**

Our results revealed that the total gas production, CH_4_ production, and dry matter digestibility reduced significantly as 3NPA doses increased in both yak and cattle. In addition, the H_2_ accumulation increased significantly in the tubes cultured with 3NPA compared to the control in both yak and cattle. The dynamic profile of total volatile fatty acid (TVFA) production, acetate concentration, and propionate concentration in both yak and cattle at 12 and 24 h incubation was consistent, in which they decreased as 3NPA doses increased, while those for 48 and 72 h incubation were divergent between them. These data demonstrated that the effects of 3NPA on fermentation characteristics between yak and cattle were divergent, and these effects were dose-dependent, and 3NPA regarded as a potential additive to mitigate methane production; although, it inhibits the dry matter digestibility in vitro, which is beneficial to determine the effective and safe dose for use to improve animal productivity in vivo.

**Abstract:**

3-nitropropionic acid (3NPA) has been proposed as an useful modifier to mitigate ruminal enteric methane emissions. However, few studies investigated the effects of 3NPA on ruminal fermentation characteristics of grazing ruminants in vitro. Rumen fluid from grazing yak and cattle were collected and incubated with additions of 0, 8, and 16 mM 3NPA. The total gas production, CH_4_ production, and dry matter digestibility significantly decreased with increasing 3NPA doses in both ruminant species (*p* < 0.05) and methane production decreased to almost 100% in cattle at 8 mM NPA but not yak, while H_2_ accumulation showed an opposite trend. The total fatty acid (TVFA) production, acetate concentration, and propionate concentration in cattle decreased as 3NPA doses increased at 12 and 24 h incubation. For yak, the H_2_ accumulation reached its apex at 8 mM NPA (*p* < 0.05). The TVFA in yak decreased significantly with increasing 3NPA doses at 12 and 72 h incubation. Moreover, the acetate concentration and propionate concentration in yak decreased as 3NPA doses increased at 12 and 24 h incubation. Overall, these findings demonstrated that 3NPA could be used as a strategy to mitigate methane emissions; although, it negatively affected the dry matter degradability in vitro.

## 1. Introduction

Methane produced by ruminants results in a series of environmental problems because it is an important anthropogenic greenhouse gas and is 28 times more potent than CO_2_ in causing global warming [1]. In addition, methane emission from rumen results in a direct energy loss between 2 and 12% for ruminants fed on different feedstuffs [2]. Although enteric methane is unfavorable environmentally and economically in ruminant production chain, methanogenesis performs a very important ecological function in maintaining low hydrogen concentrations in the rumen [3] that otherwise would inhibit rumen fermentation via disturbing the oxidation of NADH [4].

Numerous strategies have been proposed to mitigate methane emissions. For example, diet modifications, genetic selection of high feed efficient animals, and the application of feed additives could decrease methane production [5,6]. In addition, it was found that the addition of ruminal fermentation modifiers including tannins, saponins, halogenated chemical compounds, essential oils, and ionophore can reduce CH_4_ emissions remarkedly [3]. However, these strategies not only increase the cost of animal husbandry but decrease the abundances of cellulolytic bacteria [7,8], lead to intoxication [9], and the efficacy of them varies across studies [10,11].

On the other hand, nitro-compounds, such as nitroethane (NE), 2-nitroethanol (2NEOH), 2-nitro-1-propanol (2NPOH), 3-nitrooxypropanol (3NOP), 3-nitro-1-propanol (3NPOH), and 3-nitro-1-propionic acid (3NPA), have received attention as they could decrease methane production up to 90% in vitro and 69% in vivo [2]. However, NE, 2NEOH, and 2NPOH have no nutritional value for ruminants [3]. Furthermore, in some cases, supplementation of these compounds results in the reduction of feed intake [12,13] and represents a high cost [9]. In addition, most of them except for 3NPOH and 3NPA are synthetic compounds [3] and whether there is residue in animal products that may raise food safety issues is not assured. By comparison, some leguminous plants (i.e., *Astragalus*) and fungi (*Aspergillus*, *Penicillium*, and *Arthrinium*) [14] contain 3NPA that can be converted to products (b-alanine and 3-amino-1-propanol, respectively) by rumen microbiota and regarded as carbon, nitrogen, and energy by the host [15]. In addition, some forage species containing 3NPA could be used to feed ruminants safely after hay; although, it is toxic to the livestock [16]. Additionally, 3NPA is less toxic than NPOH, and the ruminants grazing on the Qinghai–Tibetan plateau could consume a moderate number of poisonous plants without obvious poisonous symptoms [17]. More importantly, it consumes the electrons available for methanogenesis and consequently reduces methane production [4]. However, how the 3NPA affects methane emissions, ruminal fermentative profiles, and forage digestibility remains unclear, especially for yak and cattle grazed on the Qinghai–Tibetan Plateau, considering that various poisonous plants such as *Oxytropis* and *Astragalus* contain 3NPA here [17]. Moreover, yak have developed a series of unique morphological and physiological mechanisms that enable them to better adapt to the harsh environment than cattle [18]. Therefore, we hypothesized that 3NPA could reduce the methane emissions significantly in both yak and cattle, but the effects of it on ruminal fermentation profiles in vitro are divergent between them. Thus, the objectives of this study were to investigate effects of different doses of 3NPA on in vitro ruminal CH_4_ production and fermentation characteristics, as well as to determine whether these effects are divergent between yak and cattle.

## 2. Materials and Methods

### 2.1. Treatment and Incubation

The experimental protocol was approved by the Animal Ethics Committee of the Chinese Academy of Lanzhou University (permission no. SCXK Gan 20140215). This study was conducted at Wushaoling in the Qinghai–Tibetan autonomous county of Tianzhu, Gansu Province (37°12.4′ N, 102°51.7′ E, 3154 m a.s.l). Rumen fluid samples were collected using a vacuum pump before morning grazing from four cattle (Qaidam, 220 ± 20 kg body weight) and four yaks (200 ± 10 kg body weight) that fully grazed on the same pasture together (Table 1). Rumen fluid from each donor was filtered via four layers of cheesecloth while maintained at 39 °C under anaerobic conditions by continuous CO_2_ flushing. The collected rumen fluid from each donor was mixed in equal proportions and then diluted in a 1:2 (*v*/*v*) with a prewarmed buffer solution to obtain the inoculum as described by Kh et al. [19]. The buffer solution mainly contained phosphate (Na_2_HPO_4_ and KH_2_PO_4_) and carbonate (NaHCO_3_ and NH_4_HCO_3_), as described by Kang et al. [20]. Afterwards, a 40 mL subsample of the inoculum was infused into a 100 mL Menke fermenter (Model Fortuna, Haberle Labortechnik, Lonsee, Germany), containing 400 mg (±5 mg) of substrate (oat hay; Table 1). The substrate (grounded through a 1 mm-sieve) was placed into the nylon bags and then glass beads were poured into the bags to avoid surface floating [21]. Next, the nylon bags (54 um) were sealed and put into the Menke fermenter. The cultures swirled gently and incubated at 39 °C for 72 h. Three 3-nitropropionic acid (3NPA) dose levels (0, 8, and 16 mmol/L), which were obtained by adding 0, 0.95, and 1.90 g NPA (Sigma-Aldrich, Milwaukee, WI, USA) to the inoculum per litter, respectively, used in the trail. The experiment was conducted in two consecutive runs with 4 replicates of each time point within each treatment (3 blanks for each treatment within each run).

### 2.2. Sampling and Measurements

Gas production was recorded during 12, 24, 48 and 72 h incubation. Headspace gas (5 mL per fermenter) was collected from four fermenters using a sealed gas injection needle. Concentration of CH_4_ and H_2_ in the samples were determined using a gas chromatograph (SP-3420A type, Beifeng-Ruili Analytical Instrument Co. Ltd., Beijing, China), and the tested equipment settings and conditions followed Wang et al. [22].

After 12, 24, 48 and 72 h incubation, four fermenters were placed into the ice water to terminate the incubation followed by measuring the pH using a portable pH meter (Sartorius PB-10, Goettingen, Germany). In addition, the liquid sample was collected from four fermenters and frozen in liquid nitrogen and then stored at −80 °C for VFA. The nylon bags were washed with distilled water until the effluent was clear. After that, the nylon bags were oven dried at 55 °C for 48 h for determination of the dry matter digestibility (DMD). The volatile fatty acid (VFA) concentrations were measured by gas chromatography (SP-3420A type, Beifeng-Ruili Analytical Instrument Co. Ltd., Beijing, China) following the method described by Zhang et al. [23].

The chemical composition of oat hay and pasture, including (dry matter) DM, crude protein (CP), and ether extract (EE), were analyzed following the AOAC (2016) procedure [24]. The NDF and ADF contents in the oat hay and pasture were measured according to the method described by Van Soest et al. [25]. The dry matter digestibility (DMD) in vitro fermentation was obtained by subtracting DM residues from their initial DM amounts prior to incubation.

### 2.3. Statistical Analysis

A two-way ANOVA was applied to test the fixed effects of treatment and species with their interactions and the random effect of subjects in R 3.5.3 software. Data are presented as mean ± standard error (SEM). *p* < 0.05 was regarded as a significant difference.

## 3. Results

### 3.1. Effects of 3NPA on Ruminal pH and VFA Concentration

The effect of the interaction between species and doses were not significant for pH except for 24 h incubation, where the mean pH showed a significant difference between yak and cattle (*p* < 0.05, Table 2). An interaction between species and dose was evident for VFA concentration (mmol/L) in both yak and cattle during 72 h incubation (*p* < 0.05, Table 2). Specifically, the TVFA (total volatile fatty acid) concentration (mmol/L) decreased significantly with increasing 3NPA doses in cattle at 12 and 24 h incubation (*p* < 0.05), whereas it showed an increased trend at 48 and 72 h incubation (Table 2). Similarly, acetate and propionate concentrations decreased with the supplementation of 3NPA in cattle at 12 and 24 h incubation but increased at 48 and 72 h incubation (*p* < 0.05, Table 2). The butyrate concentration in cattle increased from 0 to 8 mM 3NPA in all cultures except for 48 h incubation where its concentration showed an opposite trend (*p* < 0.05, Table 2). The acetate to propionate ratio (A:P) increased significantly as 3NPA doses increased in cattle at 12, 24, and 48 h incubation but decreased from 8 to 16 mM 3NPA at 72 h incubation (*p* < 0.05, Table 2).

For yak, the TVFA concentration decreased significantly with the supplementation of 3NPA at 12 and 72 h incubation, while its concentration reached its apex at 8 mM at 24 and 48 h incubation (*p* < 0.05, Table 2). Moreover, the acetate concentration in yak decreased significantly with increasing 3NPA doses at 12, 24, and 72 h incubation and fluctuated at 48 h incubation (*p* < 0.05, Table 2). The propionate concentration in yak reached its apex at 8 mM 3NPA at 24, 48, and 72 h incubation but decreased greatly with the supplementation of 3NPA at 12 h incubation (*p* < 0.05, Table 2). Notably, the butyrate concentration in yak reached its peak at 8 mM 3NPA in all cultures, and the A:P ratio decreased greatly as 3NPA doses increased at 48 and 72 h incubation (*p* < 0.05, Table 2). Compared to cattle, the TVFA and butyrate concentrations in yak were greater in all cultures except for those at 48 h incubation (*p* < 0.05, Table 2).

### 3.2. Effects of 3NPA on Ruminal Dry Matter Digestibility, Total Gas, CH_4,_ and H_2_ Production

The interaction between dose and species for dry matter digestibility (DMD) did not show a significant difference except for at 72 h incubation. The DMD in the control was greater compared to the treatment cultures in both yak and cattle except for that at 12 incubation (*p* < 0.05, Table 3). For example, the DMD in cattle decreased significantly with the supplementation of 3NPA at 24 and 72 h incubation and fluctuated at 12 and 48 h incubation (*p* < 0.05, Table 3). The total gas production (mL/g of degraded DM) was affected by species and dose (*p* < 0.05, Table 3). For instance, it decreased significantly with increasing 3NPA doses in cattle at 12, 24, and 48 h incubation and reached its lowest value for 8 mM 3NPA at 72 h incubation (*p* < 0.05, Table 3). The H_2_ accumulation (mL/g of degraded DM) was affected by species and dose at 48 and 72 h incubation, and it increased significantly as 3NPA doses increased in cattle at 12 and 72 h incubation and reached its peak at 8 mM 3NPA at 24 and 48 h incubation (*p* < 0.05, Table 3). In the case of CH_4_ production (mL/g of degraded DM), the interaction between species and dose showed a significant difference at 12, 24, and 72 h incubation, and it reduced remarkably with the supplementation of 3NPA in cattle during the incubation period and even disappeared at the dose of 16 mM 3NPA (*p* < 0.05, Table 3).

For yak, the DMD decreased significantly as 3NPA doses increased at 24, 48, and 72 h incubation (*p* < 0.05, Table 3), and the total gas production (mL/g of degraded DM) decreased significantly with the supplementation of 3NPA in all cultures (*p* < 0.05, Table 3). The H_2_ accumulation (mL/g of degraded DM) increased greatly as 3NPA doses increased at 12 and 72 h incubation and reached its peak at 8 mM 3NPA at 24 and 48 h incubation (*p* < 0.05, Table 3). In the case of CH_4_ production (mL/g of degraded DM), it increased from 0 to 8 mM 3NPA and then decreased afterwards at 12, 24, and 48 h incubation, while it reduced remarkably with increasing 3NPA doses at 72 h incubation (*p* < 0.05, Table 3). For comparison between yak and cattle, the total gas production in yak was greater than cattle at each 3NPA dose at 12, 24, and 48 h incubation (*p* < 0.05, Table 3). The CH_4_ production in yak was greater than cattle in all cultures at each incubation timepoint within each dose of 3NPA (*p* < 0.05, Table 3).

## 4. Discussion

Although a variety of strategies have been proposed to mitigate methane emissions, there is still a lack of effective solutions for livestock husbandry. Therefore, 3NPA, which is naturally produced by some plants (*Astragalus*) and fungi (*Aspergillus* and *Penicillium*) [14], was tested for its dose effects on methane emissions, ruminal volatile fatty acid, and dry matter digestibility in both grazing yak and grazing cattle.

### 4.1. Effect of 3NPA on In Vitro Rumen Gas Production and Dry Matter Digestibility of Grazing Yak

Total gas production measured during 72 h incubation was reduced remarkably in yak with the supplementation of 3NPA, which agrees with the finding of previous studies where the total gas production was less in treatment groups than the control group [3,8,16], suggesting that supplementation of 3NPA inhibited the rumen fermentation, which is further supported by the decreased DMD with increasing 3NPA doses in the current study. However, it was reported that the effect of 3NPA on total gas production was modest or even had no effect [26]. This discrepancy may be ascribed to the different doses of 3NPA used among these studies as the effect of 3NPA on total gas production was dose-dependent [3]. Moreover, in contrast to the prior studies reported that the CH_4_ production reduced significantly with the supplementation of 3NPA [3,16,27], CH_4_ production showed an increased trend from 0 to 8 mM 3NPA for 12, 24, and 48 h incubation in this study. This finding suggests that the effect of 3NPA on inhibition of ruminal methane production is divergent between grazing yak and other ruminant species regarding the animal species affects ruminal fermentation characteristics [28], and further studies based on metagenome and metabolome could be conducted to dig out the potential mechanism behind this.

TVFA in yak decreased with increasing 3NPA doses, which resembles results reported by Ochoa-García et al. [3] where TVFA production of crossbreed Angus × Hereford heifers decreased with the supplementation of 3NPA; although, fluctuations were observed. Consistent with the results in yak in the present study, previous studies have found that supplementation of 3NPA can reduce the acetate concentration [3,8], which may be attributed to the reduction in dry matter digestibility. In addition, the butyrate concentration was greater in the treated cultures than untreated controls irrespective of a fluctuation at 72 h incubation, indicating that the supplementation of 3NPA facilitates the absorption of nutrients, especially VFA, given that butyrate plays an important role in energy metabolism [29].

### 4.2. Effect of 3NPA on In Vitro Rumen Fermentation Characteristics of Grazing Cattle

Regarding cattle, the DMD decreased as 3NPA doses increased, and the H_2_ accumulation was greater in all cultures compared to the control. Rumen H_2_ accumulation results in the reduction of rumen fermentation and feed degradation [30]. Therefore, reduced DM digestibility with increased accumulation of H_2_ could be expected. Moreover, total gas production decreased with the supplementation of 3NPA in all cultures but not for 72 h incubation. This finding suggests that the effect of 3NPA on gas production may be dose-dependent [3].

The novel finding in this study is that supplementation of 3NPA caused an increase in TVFA, acetate, and propionate concentrations in cattle at 48 and 72 h incubation but decreased at 12 and 24 h incubation; although, the dry matter digestibility decreased during this period. Prior studies have shown that 3NPA was rapidly degraded before 48 h incubation [3,17]. Moreover, ruminal microbiota could cleave 3NPA to nitrite [31] and high accumulations of nitrate could inhibit microbiota that are important for rumen digestion [32]. Thus, a possible reason for decreased TVFA, acetate, and propionate concentrations before 48 h incubation could be attributed to high accumulation of 3NPA metabolite (i.e., nitrite). The potential explanation for the increased TVFA, acetate, and propionate concentrations after 48 h incubation may be ascribed to the production of microbial extracellular enzymes that hydrolyze the DM into other products but VFA [33]. Notably, the acetate to propionate ratio increased with increasing 3NPA doses in all cultures except for 72 h incubation. Reducing the ruminal acetate to propionate ratio (A:P ratio) positively associated with the efficiency of dietary energy utilization [34], and thus, a reduction in dry matter digestibility with an increase in the A:P ratio could be reasonable. Consistent with the results in cattle in the present study, a previous study reported that supplementation of 3NPA increased the propionate concentration compared to the control [8]. This may be attributed to the partial conversion of 3NPA into propionic acid in the rumen [35].

### 4.3. Comparison between Yak and Cattle in In Vitro Rumen Fermentation Characteristics with 3NPA Treatment

Notably, the total gas production was greater in yak than cattle for the control. Earlier studies have found that yak has higher digestibility compared to cattle in vivo and in vitro [36,37], which contributes to numerically higher total gas production. Furthermore, the H_2_ accumulation in yak reduced remarkably when the dose of 3NPA was higher than 8 mM, whereas that in cattle increased significantly with increasing 3NPA doses. This result suggests that the hydrogen disposal pathways are divergent between yak and cattle, or yak has higher 3NPA tolerance. However, the hydrogen metabolism and methanogenesis pathways were lacking in the current study. Ecological and functional research employing metagenome and metabolome approaches on rumen microbiome and metabolites is needed to determine if unique and efficient hydrogen disposal pathways are exhibited in yak. In addition, TVFA in cattle at 72 h incubation in the present study increased with the supplementation of 3NPA, while that in yak showed an opposite trend, indicating different responses between yak and cattle to 3NPA treatment. The potential reason for this could be attributed to the different pattern of H_2_ accumulation between yak (reached its peak for 8 mM) and cattle (increased with the supplementation of 3NPA) because H_2_ accumulation always compensates by increased propionate concentration [27], which contributes to the increased TVFA production. Further studies based on metagenome and metabolome should be performed to explore the metabolism mechanisms of 3NPA in the rumen of grazing ruminants.

In contrast, the CH_4_ production was reduced significantly with increasing 3NPA doses in both yak and cattle. Similar to the results of the current study, several prior studies found that methane production markedly reduced with the supplementation of 3NPA [3,8,16,27]. Supplementation of 3NPA could inhibit methanogens directly [38] or competing with the electrons available for methanogens via suppressing ability of formate dehydrogenase in the rumen [33,34,35,36,37,38,39], which could be an explanation for this phenomenon; although, there was no information pertaining to the rumen methanogens composition and function in the current study. In addition, H_2_ accumulation in the cultures treated with 8 and 16 mM 3NPA in both yak and cattle were greater compared to the control (0 mM). This could be attributed to the lower methane production in the treated cultures than the untreated because hydrogen was supplied for methane production in the rumen [40], or the archaeal methanogenesis pathways was inhibited by the supplementation of 3NPA [41]. This observation was supported by the in vitro finding reported by Ochoa-García et al. [3], who reported that supplementation of 3NPA increased the accumulation of H_2_. Similarly, according to Ochoa-García et al. [8], hydrogen produced for the treatment tubes (3NPA) was slightly greater compared to the control. However, it was found that hydrogen accumulation was unaffected by the supplementation of 3NPA under in vitro conditions [16]. Variations in the concentration of 3NPA [3] and fermentation substrate [27] may contribute to this discrepancy.

## 5. Conclusions

The addition of 3NPA not only significantly decreased CH_4_ production but also decreased fermentation as reflected by decreased total gas production and dry matter digestibility, and its effect was dose-dependent. Moreover, the inhibition effect of 3NPA on methane emission was stronger in cattle than yak as methane production decreased to almost 100% in cattle at 8 mM NPA but not yak. In addition, the effects of 3NPA on TVFA production, acetate concentration, and propionate concentration in both yak and cattle were affected by incubation time and divergent between yak and cattle. Additionally, the effect of 3NPA on butyrate production was consistent between yak and cattle where its concentration reached apex at 8 mM 3NPA in all cultures. Future studies based on metagenomic and metabolomic approaches are needed to understand the effects of 3NPA on the rumen microbiome composition and function, which contributes to better explain how 3NPA affects rumen fermentation, as well as to determine the effective and safe dose for use to improve animal productivity.

## Figures and Tables

**Table 1 animals-14-01804-t001:** Chemical composition of experimental forage (dry matter basis).

Item	Oat Hay	Pasture
DM	94.07	93.93
CP	7.85	7.08
NDF	53.47	63.11
ADF	32.53	35.74
ADL	6.45	7.82
EE	2.16	1.25
Ash	11.15	8.32

Note: DM: dry matter; CP: crude protein; NDF: neutral detergent fiber; ADF: acid detergent fiber; ADL: acid detergent lignin; EE: Ether extract.

**Table 2 animals-14-01804-t002:** Effect of 3-nitropropionic acid on in vitro ruminal pH and VFA profile (mmol/L) after 72 h incubation.

Item	Incubation Time/h	3-Nitropropionic Acid Dose (mmol/L)						SEM	*p*-Value		
		Cattle			Yak				Dose	Species	D×S
		0	8	16	0	8	16				
pH	12	7.13	7.11	7.09	7.05	7.07	7.09	0.01	0.12	0.08	0.07
	24	7.12	7.09	7.08	7.05	7.03	7.02	0.02	0.05	0.04	0.06
	48	7.06	7.02	7.04	7.02	7.02	7.03	0.01	0.2	0.1	0.06
	72	7.05	7.01	6.98	6.95	6.98	7.01	0.01	0.1	0.07	0.06
Total VFA	12	30.57 ^a^	25.19 ^b^	14.16 ^c^	40.87 ^a^	30.17 ^b^	28.68 ^b^	1.97	<0.01	<0.01	0.1
	24	53.48 ^a^	40.47 ^b^	34.81 ^c^	57.81 ^a^	60.22 ^a^	49.14 ^b^	2.17	<0.01	<0.01	0.02
	48	76.4	90.16	89.97	70.34	83.52	70.74	2.08	0.06	<0.01	0.09
	72	87.71	94.96	97.36	106.9	102.1	90.1	1.82	0.14	0.04	<0.01
Acetate	12	20.97 ^a^	19.03 ^a^	11.17 ^b^	28.93 ^a^	21.56 ^b^	20.72 ^b^	1.29	<0.01	<0.01	0.38
	24	36.69 ^a^	29.15 ^b^	25.86 ^c^	39.85 ^a^	37.27 ^b^	34.11 ^c^	1.17	<0.01	<0.01	<0.01
	48	45.24	57.94	57.89	47.01	51.24	40.07	1.64	0.2	<0.01	<0.01
	72	54.17 ^b^	58.98 ^a^	61.14 ^a^	70.69 ^a^	62.34 ^a^	52.68 ^b^	1.33	0.09	0.07	<0.01
Propionate	12	7.13 ^a^	2.76 ^b^	1.1 ^c^	8.16 ^a^	3.64 ^b^	3.91 ^b^	0.59	<0.01	<0.01	0.2
	24	13.4 ^a^	7.11 ^b^	5.17 ^b^	12.42 ^a^	14.12 ^a^	8.81 ^b^	0.79	<0.01	<0.01	0.03
	48	23.55	25.27	24.62	16.35 ^b^	24.19 ^a^	23.14 ^a^	0.72	<0.01	<0.01	0.02
	72	25.58 ^b^	27.62 ^ab^	29.67 ^a^	25.92 ^b^	29.42 ^a^	27.3 ^ab^	0.42	<0.01	0.9	0.02
Butyrate	12	2.48	3.13	1.89	3.6	4.96	4.06	0.25	0.65	<0.01	0.1
	24	3.39	4.21	4.05	5.54	8.83	6.23	0.43	0.33	<0.01	0.97
	48	7.61	6.96	7.46	6.98	8.09	7.66	0.13	0.4	0.4	0.24
	72	7.96 ^ab^	8.36 ^a^	6.55 ^b^	9.93 ^ab^	10.4 ^a^	9.29 ^b^	0.27	0.01	<0.01	0.61
A:P	12	2.94 ^c^	7 ^b^	10.1 ^a^	3.55 ^b^	5.92 ^a^	5.29 ^a^	0.6	<0.01	<0.01	<0.01
	24	2.74 ^b^	4.1 ^a^	4.95 ^a^	3.21 ^a^	2.64 ^b^	3.87 ^a^	0.19	<0.01	<0.01	<0.01
	48	1.92 ^b^	2.29 ^a^	2.35 ^a^	2.88 ^a^	2.12 ^b^	1.73 ^c^	0.09	<0.01	0.6	<0.01
	72	2.12	2.13	2.06	2.68 ^a^	2.1 ^b^	1.93 ^c^	0.04	<0.01	0.02	<0.01

Note: Lowercase letters (a/b/c) indicate differences among treatment groups (*p* < 0.05).

**Table 3 animals-14-01804-t003:** Effect of 3-nitropropionic acid on in vitro ruminal dry matter digestibility (DMD), total gas, CH_4_, and H_2_ production after 72 h incubation.

Item	Incubation Time/h	3-Nitropropionic Acid Dose (mmol/L)						SEM	*p*-Value		
		Cattle			Yak				Dose	Species	D×S
		0	8	16	0	8	16				
DMD, %	12	39.2	39.8	37.7	38.0	36.6	37.3	0.33	0.11	0.01	0.54
	24	41.9	40.1	39.9	42.4 ^a^	40.5 ^ab^	39.5 ^b^	0.42	0.02	0.81	0.66
	48	50.8 ^a^	43.2 ^b^	44.9 ^b^	51.8 ^a^	44.5 ^b^	43.1 ^b^	0.89	<0.01	0.87	0.33
	72	59.2 ^a^	46.2 ^b^	45.9 ^b^	59.4 ^a^	50.1 ^b^	46.2 ^c^	1.45	<0.01	0.01	0.01
Total gas/mL, g of degraded DM	12	193 ^a^	185 ^a^	150 ^b^	216 ^a^	171 ^b^	157 ^b^	9.99	<0.01	0.74	0.45
	24	253 ^a^	216 ^b^	185 ^b^	394 ^a^	266 ^b^	196 ^c^	31.16	<0.01	<0.01	<0.01
	48	399 ^a^	287 ^b^	263 ^b^	507 ^a^	364 ^b^	280 ^c^	38.19	<0.01	<0.01	0.03
	72	448 ^a^	375 ^b^	402 ^b^	496 ^a^	395 ^b^	357 ^c^	11.42	<0.01	0.03	<0.01
H_2_/mL, g of degraded DM	12	3.46 ^b^	8.35 ^a^	9.11 ^a^	0.09 ^b^	8.48 ^a^	5.66 ^a^	1.45	<0.01	<0.01	0.99
	24	0.04 ^b^	2.13 ^a^	1.78 ^a^	0.07 ^c^	1.47 ^ab^	1.03 ^b^	0.36	<0.01	0.04	0.14
	48	0.02 ^b^	1.35 ^a^	1.03 ^a^	0.02 ^b^	0.28 ^a^	0.17 ^a^	0.23	<0.01	<0.01	<0.01
	72	0.01 ^b^	9.61 ^a^	9.95 ^a^	0.16 ^b^	7.32 ^a^	6.15 ^a^	0.99	<0.01	<0.01	<0.01
CH_4_/mL, g of degraded DM	12	16.3 ^a^	0 ^b^	0 ^b^	30.4 ^a^	31.3 ^a^	0.84 ^b^	6.15	<0.01	<0.01	0.03
	24	9.69 ^a^	0 ^b^	0 ^b^	21.6 ^a^	21.8 ^a^	0.87 ^b^	4.28	<0.01	<0.01	<0.01
	48	18.2 ^a^	0.93 ^b^	0 ^c^	20.1 ^a^	24.0 ^a^	8.34 ^b^	4.19	<0.01	<0.01	0.22
	72	11.8 ^a^	0.66 ^b^	0 ^c^	65.9 ^a^	12.6 ^b^	1.14 ^c^	5.64	<0.01	<0.01	<0.01

Note: Lowercase letters (a/b/c) indicate differences among treatment groups (*p* < 0.05).

## Data Availability

Data are contained within the article.

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
