# Peer review of "Effect of 3-Nitropropionic Acid at Different Doses on In Vitro Rumen Fermentation, Digestibility, and Methane Emissions of Grazing Yak and Cattle"

_animals, 2024, doi:10.3390/ani14121804_

Round 1

Reviewer 1 Report (Previous Reviewer 1)

Comments and Suggestions for Authors

Thank you for addressing all the concerns of the reviewers.

Author Response

Dear Reviewer 1;

Thank you so much for your approval and contribution during the review process, your comments and suggestions really help us improve our manuscript a lot, we really appreciate it.

Reviewer 2 Report (Previous Reviewer 2)

Comments and Suggestions for Authors

The authors have improved the manuscript according to my previous comments. 

Author Response

Dear Reviewer 2;
Thank you so much for your approval and contribution during the review process, your comments and suggestions really help us improve our manuscript a lot, we really appreciate it.

Reviewer 3 Report (Previous Reviewer 3)

Comments and Suggestions for Authors

Authors have improved the manuscript a lot. There are some issues in the revised parts and other.

Table 1: Is it correct that DM content of pasture is 93.9%? DM content of the pasture is usually less.

Table 2: the data in column denoted as 16 under yak are missing or incorrect and SEM values are missing in this tables.

Valerate, isobutyrate, isovalerate should be deleted as they were not present in detectable amount and you cannot perform stat analysis.

Table 3: there is no need to decimal places for values > 100. Present the data to maximum 3 significant digits.

Methane production as ml per unit of digested DM decreased in some time point compared with the earlier time point.  I would suggest to report methane production as ml.

Statistical analysis should be redone to indicate the differences among the doses and species when  interaction effect is significant as I find many variables are significant for the interaction effect, but it is unclear how the NOP affected between species and doses. Authors should take advice of a person good in statistics.

Conclusion and abstract: authors should emphasize their study that NOP has stronger action on cattle than yak as 8 mM concentration methane production decreased to almost 100% whereas it is not in case of yak.

Comments on the Quality of English Language

Minor English editing required.

Author Response

Response to Reviewers’ Comments

Reviewer 3

Point 1: Table 1: Is it correct that DM content of pasture is 93.9%? DM content of the pasture is usually less.

AU: Thank you so much for your comment, we conducted this experiment in November 2017 that belonged to dry season on the Qinghai-Tibetan Plateau, and thus the dry matter of pasture (93.93%) is relatively high. Similarly, the dry matter content of forage during dry season ranged from 93 to 96% were observed in the previous studies conducted on the Qinghai-Tibetan Plateau.

Refs:

Liang Z, Zhang J, Ahmad A A, et al. Forage lignocellulose is an important factor in driving the seasonal dynamics of rumen anaerobic fungi in grazing yak and cattle[J]. Microbiology Spectrum, 2023, 11(5): e00788-23.

Ma L, Xu S, Liu H, et al. Yak rumen microbial diversity at different forage growth stages of an alpine meadow on the Qinghai-Tibet Plateau[J]. PeerJ, 2019, 7: e7645.

Point 2: Table 2: the data in column denoted as 16 under yak are missing or incorrect and SEM values are missing in this tables. Please refer to line 153.

AU: The relevant information has been added in table 2 as suggested in the revised manuscript.

Point 3: Valerate, isobutyrate, isovalerate should be deleted as they were not present in detectable amount and you cannot perform stat analysis.

AU: Thank you so much for your comments, Valerate, isobutyrate, isovalerate have been deleted in the revised manuscript. Please refer to line 153.

Point 4: Table 3: there is no need to decimal places for values > 100. Present the data to maximum 3 significant digits.

AU: We agree with the reviewer’s suggestion, and we have deleted the decimal places for values > 100 in the revised manuscript and present data to maximum 3 significant digits in table 3. Please refer to line 183.

Point 5: Methane production as ml per unit of digested DM decreased in some time point compared with the earlier time point.  I would suggest to report methane production as ml.

AU: Thank you so much for your suggestion, here we would like to compare both the methane production efficiency and amount after the supplementation of 3NPA using this unit (mL/g of degraded DM), whereas it just could compare the amount of methane emissions using mL as unit. Also, the unit (mL/g of degraded DM) is very popular among previous studies. Thus, we chose mL/g of degraded DM as the unit of methane emission. 

Refs:

Júnior J G, Garcia J A B, Lino R A, et al. Effects of trace mineral source and exogenous enzymes on ruminal in vitro fermentation of roughage-based or concentrate-based simulated diets[J]. Animal Feed Science and Technology, 2024, 310: 115930.

Yi S, Zhang X, Chen X, et al. Fermentation of increasing ratios of grain starch and straw fiber: effects on hydrogen allocation and methanogenesis through in vitro ruminal batch culture[J]. PeerJ, 2023, 11: e15050.

Chen L, Bao X, Guo G, et al. Evaluation of gallnut tannin and Lactobacillus plantarum as natural modifiers for alfalfa silage: Ensiling characteristics, in vitro ruminal methane production, fermentation profile and microbiota[J]. Journal of Applied Microbiology, 2022, 132(2): 907-918.

Winichayakul S, Beechey-Gradwell Z, Muetzel S, et al. In vitro gas production and rumen fermentation profile of fresh and ensiled genetically modified high–metabolizable energy ryegrass[J]. Journal of Dairy Science, 2020, 103(3): 2405-2418.

Wang W, Ungerfeld E M, Degen A A, et al. Ratios of rumen inoculum from Tibetan and Small-tailed Han sheep influenced in vitro fermentation and digestibility[J]. Animal feed science and technology, 2020, 267: 114562.

Point 6: Statistical analysis should be redone to indicate the differences among the doses and species when interaction effect is significant as I find many variables are significant for the interaction effect, but it is unclear how the NOP affected between species and doses. Authors should take advice of a person good in statistics.

AU: Thank you so much for your suggestion, lower case letters (a/b/c) have been added in the revised tables to indicate differences among treatment groups within each species, and capital letters (A/B) reflects differences between yak and cattle within each treatment group (p < 0.05). The results showed the rumen fermentation parameters were affected by both 3NPA doses and species if the interaction effect is significant. Also, the statistical analysis of this study has been reviewed by the person who are good at statistical analysis in this revised version. Please refer to lines 153-155 and 183-185. 

Point 7: Conclusion and abstract: authors should emphasize their study that NOP has stronger action on cattle than yak as 8 mM concentration methane production decreased to almost 100% whereas it is not in case of yak.

AU: Thank you so much for your suggestion, we have added “Moreover, the inhibition effect of 3NPA on methane emission was stronger in cattle than yak as methane production decreased to almost 100% in cattle at 8 mM NPA but not yak” to highlight the difference between cattle and yak in the conclusion section of revised manuscript, and the sentence has been changed to “The total gas production, CH4 production, and dry matter digestibility significantly decreased with increasing 3NPA doses in both ruminant species (p < 0.05) and methane production decreased to almost 100% in cattle at 8 mM NPA but not yak, while H2 accumulation showed an opposite trend” to highlight the different effect of 3NPA on methane emission between yak and cattle. Please refer to Lines 26-29 and 301-303.

Point 8: Comments on the Quality of English Language

Minor English editing required.

AU: Thank you so much for your suggestion, the English has been polished by a native speaker in this revised version.

Reviewer 4 Report (Previous Reviewer 4)

Comments and Suggestions for Authors

The authors correctly addressed all the reviewer's suggestions. Consequently, I propose to publish the manuscript in its current form.

Author Response

Dear Reviewer 4;
Thank you so much for your approval and contribution during the review process, your comments and suggestions really help us improve our manuscript a lot, we really appreciate it. 

Round 2

Reviewer 3 Report (Previous Reviewer 3)

Comments and Suggestions for Authors

Authors have revised most of previous suggestions. I have following observations in this paper.

 1) Methane production as ml per unit of digested DM decreased in some time point compared with the earlier time point. I would suggest to report methane production as ml.

AU: Thank you so much for your suggestion, here we would like to compare both the methane production efficiency and amount after the supplementation of 3NPA using this unit (mL/g of degraded DM), whereas it just could compare the amount of methane emissions using mL as unit. Also, the unit (mL/g of degraded DM) is very popular among previous studies. Thus, we chose mL/g of degraded DM as the unit of methane emission.

Comments: Methane production ml per g of substrate is also widely used. It is up to authors to present it or not.

2) I have still concern on the statistical analysis. I am not clear how the authors explined the interaction effect versus dose effect. They are intermingled. Also, the species effect is not properly presented. When species effect is significant, it may not be required to use a letter to indicate it as there were to species. When interaction effect is significant, it should be compared among all treatments (6).

3) The description of the results is not proper as per the tables. For examples in lines 186-197, there is no mention of interaction effect. It is stated that there was liner effect, but authors did not analyse it. Some sentence is unclear as per the table results, etc.

Comments on the Quality of English Language

Minor English corrections are required.

Author Response

Point 1: Comments: Methane production ml per g of substrate is also widely used. It is up to authors to present it or not.

AU: Thank you so much for your comment, we agree with the reviewer’s point that ml per g of substrate is widely used, however, we would like to use mL/g of degraded DM to present the methane data in terms of production efficiency and amount in the current study.

Point 2: 2) I have still concern on the statistical analysis. I am not clear how the authors explined the interaction effect versus dose effect. They are intermingled. Also, the species effect is not properly presented. When species effect is significant, it may not be required to use a letter to indicate it as there were to species. When interaction effect is significant, it should be compared among all treatments (6).

AU: Thank you so much for your comments. The interaction effect means the in vitro rumen fermentation profiles were affected by both species and dose, and then we explored how different doses of 3NPA affected the ruminal profiles specifically, which is like previous studies. Also, we deleted the A/B in tables as suggested in the revised manuscript.

Refs:

Liu H, Zhou J, Degen A, et al. A comparison of average daily gain, apparent digestibilities, energy balance, rumen fermentation parameters, and serum metabolites between yaks (Bos grunniens) and Qaidam cattle (Bos taurus) consuming diets differing in energy level[J]. Animal Nutrition, 2023, 12: 77-86.

Mu Y Y, Qi W P, Zhang T, et al. Changes in rumen fermentation and bacterial community in lactating dairy cows with subacute rumen acidosis following rumen content transplantation[J]. Journal of Dairy Science, 2021, 104(10): 10780-10795.

Point 3: 3) The description of the results is not proper as per the tables. For examples in lines 186-197, there is no mention of interaction effect. It is stated that there was liner effect, but authors did not analyse it. Some sentence is unclear as per the table results, etc.

AU: Thank you so much for your comments, we have added description of the interaction effect of H2 and CH4 in the revised manuscript (lines 175-176 and 179-180), and we deleted the description of linear effect to avoid confusion throughout the result section. 

Point 4: Comments on the Quality of English Language

Minor English corrections are required.

AU: Thank you so much for your suggestion, we have modified our English throughout the manuscript to improve it in this revised version.

This manuscript is a resubmission of an earlier submission. The following is a list of the peer review reports and author responses from that submission.

Round 1

Reviewer 1 Report

Comments and Suggestions for Authors

This is a well written and interesting paper. I have only a few suggestions.

lines 68-71 I was 100% sure what you were trying to point out here can you please clarify.

Line 87-88 Please specify what species of cattle were being used.

Line 97 please add the pore size for the nylon bags.

Line  150 restate this without referring to numerical increase this give the reader the idea that it was significantly different when there was no difference.

Line 195 Remove Here and replace with Therefore.

Line 224 Please consider rewriting the beginning so it reads "A possible reason for the finding could be...."

Line 267 change could to should.

Line 136 Please state what statistical package was used.

Comments on the Quality of English Language

I was very impressed with the quality of English usually with papers for Animals most of my comments deal with the grammar.

Reviewer 2 Report

Comments and Suggestions for Authors

I want to encourage the authors to avoid using the same title words in the keywords.

Line 43: 25 or 28 times? Maybe the IPCC is the most credible reference about it. It could be helpful not to use the same citation (lines 43 and 45).

Line 54-56: With just 3 cited works is questionable to affirm that in 100% of the cases it happened.

Line 80: What about the hypothesis to be tested?

Table 1: Please include the dry matter and lignin content.

Why did the authors choose these 3NPA doses?

MCP evaluation is not clear, would like to know better the method of these cited kits.

Why were fermentation parameters (pH, ammonia, VFA) evaluated just on 72h? Certainly, the main effects could be observed at first fermentation hours (4-12h).

The statistical section needs to be improved, is not clear what was the experimental unit, the n either mathematical model considered (Fixed and random effects)

Authors are required to report factor interactions in the writing results.

During the discussion authors need to focus on discussing first the interactions detected and then the simple effects.

The figure could improve in clarity and letter size (too small). 

Reviewer 3 Report

Comments and Suggestions for Authors

This paper reports the effect of 3NPA on ruminal fermentation and methane production in cattle and yak. There are some studies on 3NPA in ruminants in details. This study compared the effect in cattle and yak, which might be useful, but the study is simple. Discussion is poor and is not acceptable in its present form. It seems authors are not highly expert in this field. Discussion needs substantial improvement as this section has incorrect explanations.

Other comments

L89-90: did both cattle and yak graze in the same pasture?

L108: Gas samples were collected and analyzed in different times, but the data are not presented. Not clear what was the utility of collection in different time, if the data are not presented in the paper.

L113: also see the comment above. Why data were not presented in different incubation time?

L136: Repeated measure analysis is not correct here as the data were not obtained from the same replicates in different time. Please check the repeated measure model assumption. Authors need to rerun the stat analysis.

All Tables: I suggest to present actual p-values instead of NS.

L195-196: Can you provide some references that 3NPA is produced by plants and fungi?

L221-223: What is the mechanisms that 3NPA increased ruminal VFA although it decreased digestibility?

L224-225: Contradictory statement here.

L228-230: It seems authors are not expert in ruminal fermentation.

L232-233: Please explain the mechanism with references.

L250-251: Please explain based on your results, not other results. In this study, DMD was not affected in control treatment.

L275-277: Some molecular mechanisms may be described here as this study was mainly on the mitigation of methane. For example, Methane mitigation in ruminants with structural analogues and other chemical compounds targeting archaeal methanogenesis pathways. https://doi.org/10.1016/j.biotechadv.2023.108268

L292-294: The conclusion is contradictory if the digestibility is decreased how could energy availability improved?

Comments on the Quality of English Language

Acceptable.

Reviewer 4 Report

Comments and Suggestions for Authors

The manuscript with the title: Effect of 3-nitropropionic Acid at Different Doses on In Vitro Rumen Fermentation, Digestibility, and Methane Emissions of Grazing Yak and Cattle - addresses an interesting and topical topic in the context where methane produced by ruminants results in a series of environmental problems because it is the second-most significant anthropogenic greenhouse gas. Even though the article is generally well written, it is limited in data, which does not provide a complete scientific explanation for the issue addressed. Also, the experiences were conducted in 2017 and are being submitted for publication in 2023? The authors published an article on the same topic in 2018 - the research was done on yak and Tibetan sheep (in much the same context) and looked at the microbial composition in the rumen. The objectives should be defined more clearly, in accordance with recent studies (2, 7, 13 - from the list of references).

L15: Abbreviations must be explained the first time they appear (DM, NDF, ADF).

L17: Please use the term 'volatile fatty acid' (VFA) - instead of 'fatty acid'. Thus they will not be confused with saturated FA (SFA), monounsaturated (MUFA), polyunsaturated (PUFA). Please make this change throughout the manuscript. At the first appearance use the abbreviation VFA.

LWhat novelty does your study bring compared to other previously conducted studies? (Ochoa-García, et al. In vitro reduction of methane production by 3-nitro-1-propionic acid is dose-dependent. J Anim Sci 2019, 97, 1317-13247.; Ochoa-García, et al. Effect of Ethyl Nitroacetate, Ethyl 2-332 Nitropropionate and 3-Nitropropionic Acid on Ruminal Fermentation Characteristics and Dry Matter Degradability under In Vitro Conditions. 2023; Correa, A.C.; et al. Effect of sole or combined administration of nitrate and 3-nitro-1-propionic acid on 350 fermentation and Salmonella survivability in alfalfa-fed rumen cultures in vitro. Bioresour Technol 2017, 229, 69–77.).

Please formulate much more clearly the objective and purpose of the research carried out. How were the two doses of 3NPA tested.

L32: Please rephrase according to the data in table 3 (p ˃ 0.05).

L36: Please rephrase - 3NPA could be used as an alternative (alternative to what?) (3NPA could be used as an strategy to .......).

L48: Please explain all abbreviations the first time they appear. Apply this recommendation to the entire manuscript.

L85: I don't understand why it took over 5 years to publish the results of the 2017 experiment.

L107: In the Material and methods chapter - details regarding sampling, the number of samples are missing, and the analysis methods are not properly described (they should be described briefly).

L144, 152, 154: Why don't you use the term digestibility? Why do you use the terms of: Disappearance of......; degradability of ...; apparent disappearance (for in vivo tests - the term apparent is ok). Please use unitary terms.

L161: acetate to propionate ratio did not change in cattle (p˃0.05) (table 5). Please correct (rephrase).

L294-296: Please revise this sentence: the acetate:propionate ratio did not decrease in cattle - I don't understand how a decrease in the acetate:propionate ratio leads to increased energy production and availability.